# Validity and Reliability of Kinvent Plates for Assessing Single Leg Static and Dynamic Balance in the Field

**DOI:** 10.3390/s23042354

**Published:** 2023-02-20

**Authors:** Hugo Meras Serrano, Denis Mottet, Kevin Caillaud

**Affiliations:** 1Science Department, Kinvent, 34000 Montpellier, France; 2EuroMov Digital Health in Motion, Univ Montpellier, IMT Mine Alès, 34090 Montpellier, France

**Keywords:** validity, reliability, unipodal balance, centre of pressure, force platform

## Abstract

The objective of this study was to validate PLATES for assessing unipodal balance in the field, for example, to monitor ankle instabilities in athletes or patients. PLATES is a pair of lightweight, connected force platforms that measure only vertical forces. In 14 healthy women, we measured ground reaction forces during Single Leg Balance and Single Leg Landing tests, first under laboratory conditions (with PLATES and with a 6-DOF reference force platform), then during a second test session in the field (with PLATES). We found that for these simple unipodal balance tests, PLATES was reliable in the laboratory and in the field: PLATES gives results comparable with those of a reference force platform with 6-DOF for the key variables in the tests (i.e., Mean Velocity of the Center of Pressure and Time to Stabilization). We conclude that health professionals, physical trainers, and researchers can use PLATES to conduct Single Leg Balance and Single Leg Landing tests in the laboratory and in the field.

## 1. Introduction

Postural balance allows the maintenance of a specific posture in reaction to an external disturbance. A malfunction of any of the three physiological sensory systems (visual, vestibular, and proprioceptive) at the root of this regulation can cause imbalance problems [1]. Therefore, clinical tests are essential to identify certain issues in order to prevent fall problems [2], injuries, and a lack of stability in certain joints [3,4,5]. Postural balance tests are very useful to monitor ankle instabilities in athletes or patients [6,7].

Postural control is classically assessed by bipodal and unipodal static or dynamic balance tests [7,8,9]. These can be performed either (1) subjectively by means of an error score recorded by sports professionals or physiotherapists [5] or (2) objectively by means of force platforms [10]. Balance assessment using Center of Pressure (CoP)-derived parameters appears effective for estimating fall risk in at-risk individuals such as the elderly [10] or people with motor disorders [11] and joint instability, but also in athletes, sportsmen [12], and non-athletes [3]. CoP parameters quantified and assessed using force platforms [4,12,13,14,15] are used as a gold standard measure of balance [16].

The parameters typically used to study bipodal balance, especially to assess the risk of falling in the elderly, relate to the movement of the CoP: range of the CoP, path length, average speed, and area of the confidence ellipse [17].

To characterize the difference in bipodal balance between young and older adults with eyes open and closed, many variables related to the trajectory of the CoP can be used, either in the time domain (e.g., mean velocity, distance, or area) or in the frequency domain (e.g., total power, median frequency). However, mean CoP velocity was the measure that best identified age-related changes and differences between eye conditions in the two groups [18].

While using the Single Leg Balance test is common practice for predicting functional instability of the ankle [4] and identifying the risk of ankle sprains [5], very few studies have identified specific CoP parameters for further investigation in the analysis of unipodal postural control using force platforms. Interestingly, the CoP mean velocity appears to be the most reliable stabilometric parameter for differentiating a group of patients with a history of ankle sprains from a healthy group during a Single Leg Balance with eyes closed [4]. In addition, this variable was also the most reproducible parameter in patients with an anterior cruciate ligament rupture who performed an open-eye unipodal balance test [13]. Comparing a Single Leg Balance test, or other balance exercises, with eyes open and closed allows the quality of sensory integration of information to be assessed [19].

Another test for balance evaluation, the Single Leg Landing, is of great interest for predicting injuries of the anterior cruciate ligament [3] or ankle sprains [12]. During Single Leg Landing, the stabilization time of the ground reaction force along the anteroposterior and mediolateral axes seems moderately accurate in distinguishing stable from unstable ankles [20].

However, analyzing human movement outside of research laboratories or hospital settings is a critical issue [21]. As a result, the development of a standardized protocol for quantifying postural balance using a portable and user-friendly solution represents a clinical innovation. Indeed, it has the potential to democratize the objective assessment of balance, which has previously been primarily reserved for expert biomechanists within research institutions or universities. A lightweight pair of force platforms equipped with a unidirectional strain gauge (vertical axis) that connects via Bluetooth to a smartphone app (PLATES, Kinvent, Montpellier, France) is an intriguing alternative to the force platforms currently in use.

The objective of this study was to validate this new connected solution to assess unipodal balance in the laboratory and in the field. Our idea was that unipodal balance assessment in the field could be the tool of choice to monitor ankle instabilities in athletes or patients. We hypothesized that the postural stability measurements provided by a portable light-weight force plate system would be valid and reliable not only in the laboratory but also in the field or in clinical settings. A group of healthy female participants performed static and dynamic balance exercises on both types of force platforms to test this hypothesis.

## 2. Methods

### 2.1. Participants

Fourteen healthy and athletic women (age: 26.4 ± 7.1 years; weight: 59.5 ± 7.6 kg; height: 164.6 ± 8.1 cm; right and left ankle dorsiflexion: 37.0 ± 9.1°/35.9 ± 7.3°) volunteered to participate in the study. All participants were right-handed. They all completed an online questionnaire and reported that they had no balance problems, lower limb injuries, or back injuries. We focus on women because they are more likely than men to suffer ankle injuries [22]. Participants completed the CAIT-F (Cumberland Ankle Instability Tool—French version), which is a questionnaire to diagnose functional ankle instability [23]. If the CAIT-F score was less than 23, then they were excluded from the study.

They completed 3 sessions separated by 5 to 7 days, depending on their availability.

To accommodate individual circadian rhythms, all sessions were conducted at the same time of day. Participants were asked to maintain their eating and sleeping habits and to refrain from strenuous physical training on the day before each test. Participants were verbally encouraged throughout the tests.

All participants gave signed informed consent for inclusion before they participated in the study. The study was conducted in accordance with the Declaration of Helsinki, and the protocol was approved by the local ethics committee (IRB-EM 2201A).

### 2.2. Experimental Design

The 1st session consisted of meeting the participants at the research laboratory. Anthropometric measurements such as weight, height, and ankle range of motion (ROM) in dorsiflexion were taken. The weight-bearing lunge test was used to assess the ROM of the ankle dorsiflexion [24]. They were also familiarized with the force platforms and the exercises to be performed during the 2 future testing sessions.

For the 2nd session, before starting the tests, the participants performed a standardized warm-up that included head-to-toe joint unlocking, muscle activation, and ballistic movements of the lower limbs, followed by more specific exercises to warm up the ankles.

Participants performed the Single Leg Balance (SLB) test in several randomized conditions: (A) 3 repetitions of the right and left leg with open eyes (OE) on the PLATES; (B) 3 repetitions of the right and left leg with OE on the AMTI (reference platform); (C) 3 repetitions of the right and left leg with eyes closed (CE) on the PLATES; (D) 3 repetitions of the right and left with CE on the AMTI. The SLB test consisted of standing still on one leg [20] for 10 s while fixing a point 5 m away [25], with the hands on the hips and the non-load-bearing leg slightly bent at the hip and knee. In order to compare the results between open and closed eyes, a test duration of 10 s was chosen based on the time norms of the closed eyes condition during a unipodal balance exercise (norm = 9.4 s) [26].

A second test, the Single Leg Landing (SLL), was performed in 2 randomized conditions: (A) 3 repetitions of the right and left leg on the PLATES; (B) 3 repetitions of the right and left leg on the AMTI. This dynamic unilateral balance exercise consisted of stepping down from a step placed 19 cm above the force platform with a bounce (both feet had to be suspended before landing) [12], then stabilizing on one leg for 15 s with hands on hips and gazing at a point 5 m away. For both exercises, the trial was aborted and repeated if the participant’s non-supporting leg touched the ground, touched the supporting leg, or if the subject lost balance completely. Each trial was separated by 10 s of passive recovery, while each condition was separated by 30 s of rest.

Participants performed 3 tests per leg and per condition, and we retained the mean of the 3 tests in order to take into account the influence of the natural variability of human behavior [27]. On average, 2 trials of SLB and SLL were excluded per patient and had to be repeated.

The 3rd session was identical to the 2nd with the PLATES but did not involve the AMTI as it was performed in an ecological context outside laboratory settings. The session could take place indoors or outdoors, i.e., either in the participant’s club or association, in a classroom, or at the workplace. The warm-up was the same as in the previous test session. The participants were asked to perform the SLB and SLL tests. The same instructions and conditions were applied.

### 2.3. Materials and Measures

The tests were performed using the PLATES v3 (Kinvent, Montpellier, France) and AMTI BP400600 (AMTI, Inc., Watertown, MA, USA) force platforms. The data were collected via a Bluetooth connection by the KINVENT PHYSIO application and stored on a Samsung tablet in the case of tests with the PLATES force platforms. In the case of tests with the AMTI force platforms, the data were collected via a wired connection by the VICON Nexus software and stored on a computer.

### 2.4. Data Processing

Data were processed using Matlab R2022b software (MathWorks, Inc., Natick, MA, USA). After downsampling the AMTI recordings from 1000 Hz to 250 Hz to match the PLATES sampling rate, the AMTI and PLATES data were processed in exactly the same way. All data were filtered using a 2nd order low-pass Butterworth filter with a cut-off frequency of 10 Hz for unipodal static balance [9,28,29] and 12,53 Hz for unipodal dynamic exercise [19,29]. For each condition, we did not use a representative trial, but we computed the mean of the 3 repetitions.

For the SLB test, we focused on the Mean Velocity of the CoP global (MVcop), which is the most classical measure of interest in the assessment of the balance [13,15,20]. Other parameters such as the Path Length covered by the CoP along the anteroposterior axis (PLap) and along the mediolateral axis (PLml), the Path Length covered by the CoP global (PLcop), the Mean Velocity of the CoP along the anteroposterior axis (MVap) and along the mediolateral axis (MVml), and the surface area (SA) were also analyzed (see Appendix A).

For the SLL, we focused on the stabilization time (TTS), which represents the time it takes for an individual to return to a baseline or stable state after a Single Leg Landing. The sequential estimation method of TTS was used as described in the study by Colby et al. [3]. Other parameters such as PLcop, MVcop, and SA were also analyzed. Indeed, these CoP parameters were analyzed from the moment the patient stabilized until the end of the test (see Appendix A).

Note that we selected these metrics because they seem well adapted to investigating potential post-injury ankle or knee instability in young, healthy, active individuals [3,12]. However, due to the differences between AMTI and PLATES, we predict that MVcop would be valid (as it depends only on Fz) and that TTS would be less valid because it is better assessed with horizontal forces that PLATES do not measure.

### 2.5. Statistical Analysis

The significance level was set at *p* < 0.05.

Statistical analysis was performed using RStudio (version 2022.02.1+461) and R (version 4.1.1).

We first addressed the reliability of the PLATES under laboratory conditions (i.e., we compared the measures between AMTI and PLATES). Then, we addressed the reliability of the PLATES outside laboratory conditions (i.e., we compared the measures with the PLATES in the laboratory vs. in the field, and with the PLATES in the field vs. AMTI).

The reliability was assessed using the Bland and Altman method [30], linear regressions, and the Intraclass Correlation Coefficient (ICC 3.1).

To see if our balance data reproduced the classical results of the literature, the results obtained with the right and left leg were compared in laboratory conditions and outside laboratory conditions for SLB and SLL. The same analysis was performed to compare the eyes open and eyes closed conditions for SLB.

## 3. Results

### 3.1. Replication of Classical Results

For the Single Leg Balance test, the MVcop values obtained with PLATES under laboratory conditions were 39.1 ± 4.3 and 40.7 ± 7.4 mm/s with eyes open and 72.7 ± 10 and 78.4 ± 9.3 mm/s with eyes closed for the left and right leg, respectively. These MVcop values are similar to previous study results [31]. Concerning the results of MVcop with the PLATES under field conditions, we obtained 38.3 ± 5.7 and 41.6 ± 6.3 mm/s with eyes open and 76.3 ± 9.3 and 82.2 ± 12.3 mm/s with eyes closed for the left and right leg, respectively. To end, the MVcop values obtained with AMTI under laboratory conditions were 39.0 ± 5.5 and 41.0 ± 5.7 mm/s with eyes open and 86.8 ± 14 and 91.3 ± 16.9 mm/s with eyes closed for the left and right leg, respectively.

Comparable results for the right and left lower limbs were observed with PLATES and AMTI (Figure 1). MVcop values are significantly higher in the eyes closed condition compared with the eyes open condition (Figure 1). Indeed, regardless of the tool used, MVcop results with eyes closed are more or less two times higher than results with eyes open.

For the Single Leg Landing test, the TTS values obtained with PLATES under laboratory conditions were 2.98 ± 0.21 s for the right leg and 2.99 ± 0.17 s for the left leg. For TTS results with PLATES under field conditions, we obtained 3.03 ± 0.23 s and 2.95 ± 0.17 for the right and left legs, respectively. Finally, the TTS values obtained with AMTI under laboratory conditions were 3.24 ± 0.36 s for the right leg and 3.31 ± 0.34 for the left leg. Comparable results for the right and left lower limbs were observed, regardless of tool used.

### 3.2. Reliability in the Laboratory

PLATES-Laboratory vs. AMTI-Laboratory

In the SLB test, good reliability has been observed for the MVcop with an ICC of 0.88. The Bland and Altman plot indicated that the MVcop obtained from PLATES in the laboratory was on average 6.8  mm/s lower than the MVcop obtained from AMTI (Figure 2, top-left panel). The regression line with a slope of 0.75 showed that the underestimation is proportional to the average speed (Figure 2, top-right panel).

Concerning the SLL test, moderate reliability has been observed for the TTS variable with an ICC of 0.73. The Bland and Altman plot indicated that TTS obtained from PLATES in the laboratory was on average 0.29 s (9%) lower than the TTS obtained from AMTI (Figure 2, bottom-left panel), and the regression line with a slope of 0.66 showed that the underestimation is proportional to the average TTS (Figure 2, bottom-right panel).

For the other parameters of the SLB test, no significant differences were found between the two force platforms in the open eyes condition (see details in Appendix A). The consistency between both devices for the SLB test was good for almost all conditions (ICC > 0.75). However, this was not the case in the open eyes left leg condition for all parameters, in the closed eyes left leg condition for SA (poor), and in the right leg open eyes condition (poor). However, in the right leg with closed eyes condition, all parameters had a very good correlation (0.81 < ICC < 0.91). The bias values tended to be lower in the open eyes condition for the SLB and higher for the left leg than the right leg in the closed eyes condition. Indeed, significant bias between the two devices was observed for all parameters only in the closed eyes condition for both legs except for SA.

For the other parameters of the SLL test, the consistency between PLATES and AMTI was good for all parameters (ICC > 0.75), except for the SA (poor) and TTS in left leg condition (moderate). The bias values tended to be higher for the TTS and PLcop during the test on the left leg. Indeed, significant biases between the two devices were observed for TTS and PLcop in the left leg condition.

### 3.3. Reliability Outside Laboratory Conditions

PLATES-Field vs. PLATES-Laboratory

In the SLB test, excellent reliability has been observed for MVcop with an ICC of 0.93. The Bland and Altman plot indicated that the MVcop obtained from PLATES in a laboratory setting was on average 1.9 mm/s lower than the MVcop obtained from PLATES in a field setting (Figure 3, top-left panel), which represents a bias of about 3%. The regression line with a slope of 1.02 indicated a small bias between the two measures. (Figure 3, top-right panel).

In the SLL test, moderate reliability has been observed for the TTS variable, with an ICC of 0.72. The Bland and Altman plot indicated that the TTS obtained from PLATES in a laboratory setting was on average 0.01 s lower than the TTS obtained from PLATES in a field setting (Figure 3, bottom-left panel), which represents a bias of about 0.3%. The regression line with a slope of 0.97 indicated a small bias between the two measures. (Figure 3, bottom-right panel).

For the other parameters of the SLB test, the consistency between laboratory and field conditions was good for most parameters (see details in Appendix A). However, this was not the case in the open eyes left leg condition for PLml, MVml, and SA (moderate). Conversely, PLap, PLcop, MVap, and MVcop had good to excellent reliability in the field with the PLATES (0.80 < ICC < 0.93).

For the other parameters of the SLL test, the reliability in the field with the PLATES was good for PLcop and MVcop for both legs (>0.80). Consistency between laboratory and field measures with PLATES was good for TTS on the right leg (ICC = 0.88), moderate for TTS on the left leg (ICC = 0.63), and poor for SA (0.07 and 0.42 for the right and left leg, respectively).

b.PLATES-Field vs. AMTI-Laboratory

In the SLB test, good reliability has been observed for MVcop, with an ICC of 0.87. The Bland and Altman plot indicated that the MVcop obtained from PLATES in a field setting was on average 4.7 mm/s (9%) lower than the MVcop obtained from AMTI (Figure 4, top-left panel), and the regression line with a slope of 0.80 showed that the underestimation is proportional to the average speed (Figure 4, top-right panel).

Concerning the SLL test, good reliability has been observed for the TTS variable with an ICC of 0.73. The Bland and Altman plot indicated that TTS obtained from PLATES in a field setting was on average 0.29 s (9%) lower and significantly lower (*p* < 0.05) than the TTS obtained from AMTI (Figure 4, bottom-left panel). The regression line with a slope of 0.80 showed that the underestimation is proportional to the average speed (Figure 4, bottom-right panel).

To summarize the main reliability results, we found that in the laboratory, MVcop and TTS were reliably measured with PLATES, albeit with a proportional underestimation (Figure 2). We also found that with PLATES, MVcop was very reliably assessed in the field, but for the TTS, reliability was moderate (Figure 3). As a consequence of the previous two results, we also found that MVcop and TTS measured in the field with PLATES were reliable, although with a proportional underestimation, as in the laboratory (Figure 4).

The details of the reliability results between PLATES-Lab vs. AMTI-Lab, PLATES-Lab vs. PLATES-Field, and PLATES-Field vs. AMTI-Lab are presented in the Appendix A.

## 4. Discussion

Through this study, our results replicate classical findings in healthy individuals with the SLB test: comparable results for the right and left lower limbs and higher stability with the eyes open (Figure 1). The fact that stability decreases when eyes are closed was already reported in lab settings. Indeed, patients are two times less stable than those with their eyes open [32]. Here, we extend this finding to field settings, and this is important because it shows once again that PLATES are a reliable tool.

The replication of classical results is capital for clinicians or researchers with their patients. Indeed, some balance tests can diagnose or predict pathologies such as ankle sprains or knee injuries [3,4,20]. Moreover, indicators such as the Romberg Quotient, which is calculated as the ratio of closed eyes to open eyes, can be used to evaluate the integration of visual afferents to diagnose postural blindness, immaturity, or impairment of the role of visual and oculomotor afferents in postural regulation [32].

Next, we showed that PLATES had good reliability between PLATES-Lab and PLATES-Field and between PLATES-Lab and AMTI-Lab for the assessment of the average CoP displacement velocity during a static unipodal balance and moderate reliability for the assessment of TTS during a Single Leg Landing test.

The MVcop and TTS results showed moderate to excellent reliability between PLATES-Lab and PLATES-Fields, and the Bland and Altman plots showed a small bias (Figure 3). Moreover, our results showed that PLap, PLcop, MVap, and MVcop were the most reliable parameters for the SLB, which correlates with Li et al.’s [14] study during a bipodal balance exercise performed on a group of elderly people. SA was found to be the least reliable parameter, as the ICC varied between 0.54 and 0.87 depending on the conditions. Salavati et al. [11] had already obtained a poor and moderate ICC of SA in the eyes open and eyes closed conditions, respectively, in an assessment of bipodal balance in patients with musculoskeletal disorders.

The reliability results showed that the MVcop and TTS measured with the PLATES in a laboratory setting were strongly related to those measured using the AMTI (Figure 2).

In a laboratory and field setting, PLATES results underestimated the TTS (*p* < 0.05) compared with the AMTI, but the MVcop and TTS are still well correlated between both tools (Figure 4). A small difference between the two devices in the values of MVcop and TTS was revealed by the Bland and Altman diagrams. However, these values were consistent with the test condition for the PLATES and may have been due to the accuracy and sensitivity of the sensors or the difference in the texture and hardness of the surfaces of the platforms [33]. The significant difference in TTS for PLATES-Lab vs. AMTI-Lab and for PLATES-Field vs. AMTI-Lab may be explained by the technical execution of the reception: it is likely that participants took into account the smaller size of the PLATES when organizing their ground landing. Indeed, participants stabilized more slowly on the much larger AMTI (i.e., 600 by 400 mm vs. 330 by 175 mm).

For all comparisons, the dynamic balance test showed that TTS was a moderately reliable parameter; those results are lower than those of Colby et al. [3], who observed ICCs of 0.90 and 0.93 in SLL for the dominant and non-dominant leg, respectively. Surprisingly, in our study, the MVcop variable appeared to be the most reliable parameter for the SLL, and this has never been studied before. This highlights the potential for CoP parameters to be of interest during stabilization phases following challenging postural tasks.

Interestingly, we observed better reliability on the right side for most parameters, but only in the open eyes condition. The greater consistency on the right side is likely because side dominance impacts unipodal balance proficiency (i.e., here most participants were right-handed and right leg dominant). The fact that this is observed only in the open eyes condition suggests that unipodal balance proficiency mostly relies on visual input, which is coherent with the idea of nonlinear gains in sensorimotor integration for postural control [19].

Although the results presented are promising, this new low-cost device has limitations because PLATES are unidirectional platforms (vertical only). For example, several studies have calculated TTS not as a function of vertical force but as a function of anteroposterior or mediolateral force [20,34], which is more accurate [20]. Nevertheless, a mobile application that enables the use of a pair of portable, lightweight force PLATES and automatically calculates all CoP parameters, a task that is typically performed by specialized software and requires advanced training, would be of immense benefit to practitioners.

This study also has several limitations. First, the sample size used in the present evaluation was small, and only women participated in the study. Second, it is a predictive validity study that was conducted. Therefore, human variability between trials played an important role in the results. In addition, in the laboratory session (PLATES lab vs. AMTI), participants were asked to perform a total of 24 SLB and 12 SLL. Although the trials were randomized, fatigue could have impacted the data. Several days elapsed between the tests with the PLATES. Therefore, a little training effect could not be completely excluded.

## 5. Conclusions

Our objective was to investigate the validity and reliability of a new connected solution to assess the unipodal balance of patients in the lab and in the field.

In this paper, we showed that PLATES can be used as an alternative to AMTI (the gold standard) force plates to assess unipodal static and dynamic balance in the laboratory. Due to their ease of use, lightness, and mobility, the PLATES can be used in the field or in clinical routine for semi-automatic quantitative diagnostics guiding individualized rehabilitation of the lower limb.

In addition, the engineering of the solution has enabled the automation of data processing, the reduction of production costs, and the portability of platforms. Future research should examine the validity and reliability of these platforms on other balance exercises and on other types of populations (pathological, etc.), in order to subsequently detect and prevent injuries in sportsmen or sportswomen or for the early detection of falls in the elderly.

## Figures and Tables

**Figure 1 sensors-23-02354-f001:**
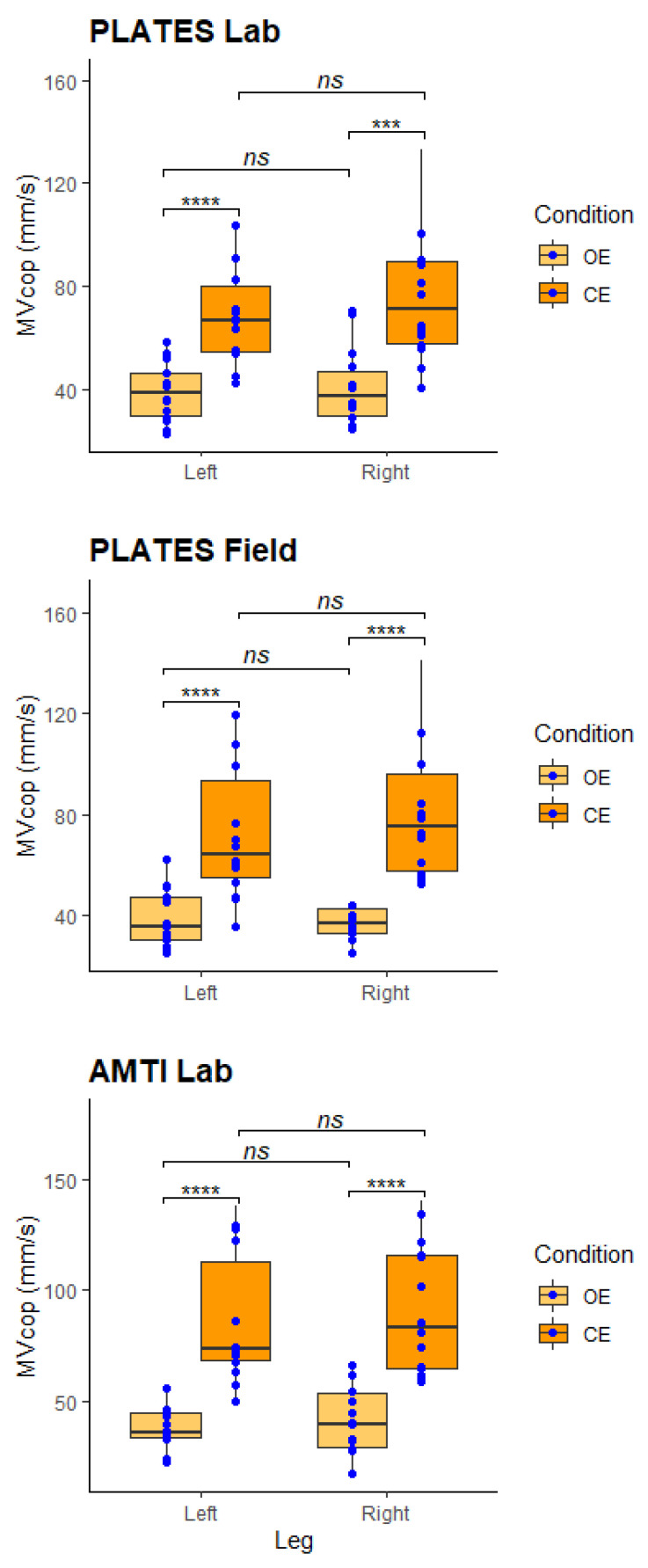
MVcop during the Single Leg Balance test. Upper panel: AMTI measures in the laboratory. Middle panel: PLATES measures in the laboratory. Lower panel: PLATES measures in the field. In each panel, the boxplot represents MVcop in the 4 experimental conditions: standing on the left/right leg with the eyes open/closed. Each blue circle represents the average of the 3 repetitions performed by each participant (*n* = 14). Paired Wilcoxon test, *ns*: no significant difference, **** p* < 0.001, ***** p* < 0.0001. The figures replicate the classical results that stability is lower with the eyes closed for both legs.

**Figure 2 sensors-23-02354-f002:**
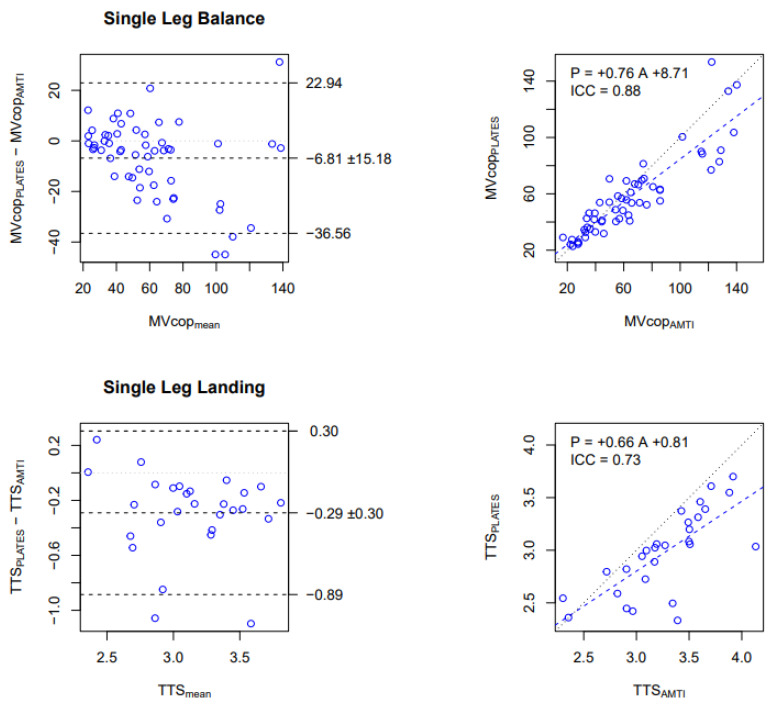
Comparison of MVcop and TTS obtained with the PLATES vs. AMTI in the Laboratory. Upper row: MVcop for the Single Leg Balance test (mm/s). Lower row: TTS for the Single Leg Landing test (s). Left column: Bland and Altman plot. The central horizontal line indicates the mean of the differences (systematic bias), which is 0 for perfect agreement. The horizontal lines above and below represent the 95% limits of agreement. Right column: Linear regression. Data from all conditions are reported (i.e., 14 participants × 4 conditions for SLB, 14 participants × 2 conditions for SLL). Each dot corresponds to the average over the 3 repetitions in each condition. The figures show that MVcop and TTS are reliably assessed in the laboratory with PLATES, although with a small underestimation (negative bias in Bland and Altman) that is proportional to the value (slope lower than 1 in the regression).

**Figure 3 sensors-23-02354-f003:**
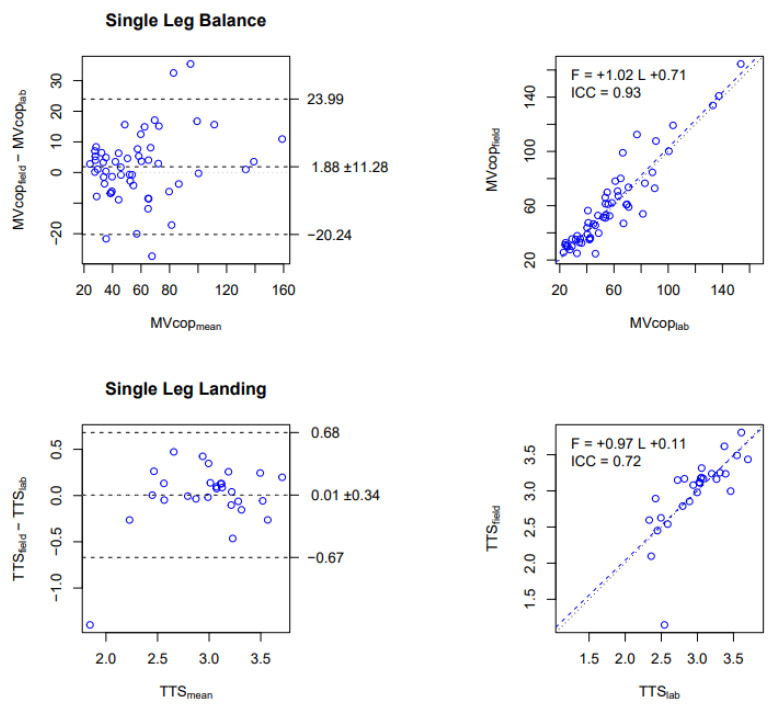
Comparison of MVcop and TTS obtained in the Laboratory vs. in the Field with the PLATES. Upper row: MVcop for the Single Leg Balance test (mm/s). Lower row: TTS for the Single Leg Landing test (s). Left column: Bland and Altman plot. The central horizontal line indicates the mean of the differences (systematic bias), which is 0 for perfect agreement. The horizontal lines above and below represent the 95% limits of agreement. Right column: Linear regression. Data from all conditions are reported (i.e., 14 participants × 4 conditions for SLB, 14 participants × 2 conditions for SLL). Each dot corresponds to the average over the 3 repetitions in each condition. The figures show that MVcop and TTS are strongly correlated between PLATES in laboratory and field settings. Indeed, negligible biases are observed in Bland and Altman diagrams that are proportional to the values (slope near 1 in the regression).

**Figure 4 sensors-23-02354-f004:**
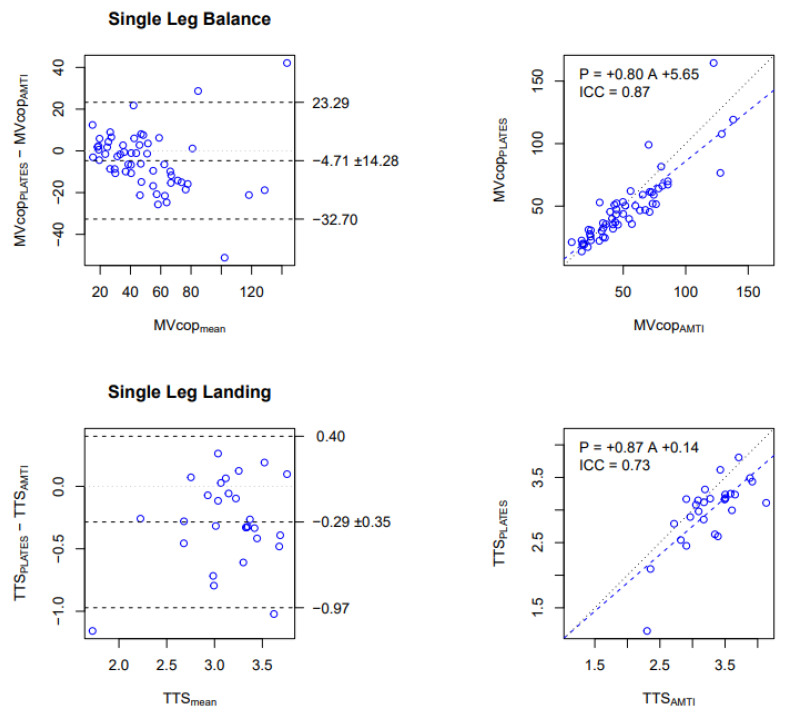
Comparison of MVcop and TTS obtained with PLATES-Field vs. AMTI-Laboratory. Upper row: MVcop for the Single Leg Balance test (mm/s). Lower row: TTS for the Single Leg Landing test (s). Left column: Bland and Altman plot. The central horizontal line indicates the mean of the differences (systematic bias), which is 0 for perfect agreement. The horizontal lines above and below represent the 95% limits of agreement. Right column: Linear regression. Data from all conditions are reported (i.e., 14 participants × 4 conditions for SLB, 14 participants × 2 conditions for SLL). Each dot corresponds to the average over the 3 repetitions in each condition. The figures show that MVcop and TTS are reliably assessed in the field with PLATES, although with a small underestimation (negative bias in Bland and Altman) that is proportional to the value (slope lower than 1 in the regression).

## Data Availability

Data are public at OSF.io: https://Osf.Io/85aqx/?View_Only=06a2a3c2a54e48718de13708fe2c20ac (accessed on 13 February 2023).

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
