# Peer review of "Validity and Reliability of Kinvent Plates for Assessing Single Leg Static and Dynamic Balance in the Field"

_sensors, 2023, doi:10.3390/s23042354_

Round 1

Reviewer 1 Report

Thank you for the opportunity to review the paper “Validity and Reliability of Kinvent Plates for assessing single leg static and dynamic balance in the field”. This study investigates whether the forceplates “PLATES” produced by a particular company can reliably assess the static one-leg balance in a) quiet single leg stance and b) one-leg landing. The study compares a) PLATES with a reference forceplate (AMTI) in a laboratory, b) PLATES in the laboratory vs PLATES in the field (participants’ homes, workplaces etc). Path length, mean velocity, surface (area of sway 95%), and time to stabilisation were assessed for both legs. Intraclass Correlation Coefficient revealed strong agreement between the laboratory and the field measurements with PLATES and fair or good agreement between PLATES and AMTI. Further, the study managed to show differences between balance with eyes open and eyes closed using PLATES, replicating a phenomenon which is well established in the literature of Motor Control.

In general, the strength of the PLATES is the lightness and the ease of use (line 334). Also, they provide values relatively coherent to the AMTI platform, and reproducible in the field. However, these conclusions are based on a couple of variables discussed in this paper. More and clinically meaningful variables and tests are sacred in order to make conclusions.

The study is well performed and well presented. There are however several limitations which have to be taken into account for the manuscript to be considered for publication.

It remains unclear how the postural variables were selected. To my humble understanding, the study aims to evaluate the suitability of PLATES for clinical assessments. If I understand correctly, the aim of the study is to assess the validity and reliability of PLATES. Experts in Biomechanics and/or clinicians using the device, might be interested in different variables as well. For example, please see the landmark study of Prieto et al 1993 (doi: 10.1109/10.532130), describing the variables that distinguish between healthy young and older adults. These include additional sway area and frequency measures. Also, it is not clear why the Anteroposterior measurements are not presented. One would expect that falling from a height should introduce instability to the AP component as well, due to the initial step-introduced momentum.

In the same vein, it is not clear why authors only present and discuss TSS and CoP velocity.

It is appreciated that authors provide evidence on the selection of the single leg tasks (lines 47-66), but double leg balance is also successful in identifying fallers and people with musculoskeletal diseases. Considering the relative danger of the single leg tasks for individuals with poor balance or suffering from ankle instability, further evidence/justification is needed to support the selection of the single leg balance tasks over the traditionally used double leg balance tasks (simple balance, Tandem stance, Romberg test etc). Please clearly state and support the task's physiological/biomechanical/functional importance.

The replication of the eyes closed/open result is an excellent idea. However, there are two points to be considered:

a)       It is mentioned at the line 105 for the first time, as coming out of the blue. I would suggest adding a rationale to support the utilisation of this test alongside hypotheses in the introduction section.

b)      Other comparisons that are well established in the literature could be performed, such as young vs old adults (or people with neurological diseases), fallers vs non-fallers, or even dominant vs non dominant limb.

Minor comments:

Tables and figures:

Table 1: I am not sure if this table adds something to the paper in its current form. Please either expand it with more bipedal balance studies or delete.

Table S1 and S3: please correct the typo in the last column (“Biais” to “Bias”)

Figures: Please explain the dashed lines (eg., 0, mean, +1 std, -1std, coef=1) in the caption

Methods:

Line 104: please state that this forceplate was used as a gold standard reference.

Lines 102-121: Please clarify what was randomised: tests, conditions, repetitions?

Lines 116, 117: It would be a good idea to write how many trials were excluded

Lines 120, 121: This statement is too vague. Did authors measure the variability? If yes, what was the criterion?

Lines 332-333: this contradicts the statement 2 paragraphs above "Because PLATES are unidirectional platforms (Fz only), they cannot replace gold-315 standard 6 DOF force platforms "

Author Response

Thank you for your comments. You will find our different answers in bold. 

It remains unclear how the postural variables were selected. To my humble understanding, the study aims to evaluate the suitability of PLATES for clinical assessments. If I understand correctly, the aim of the study is to assess the validity and reliability of PLATES. Experts in Biomechanics and/or clinicians using the device, might be interested in different variables as well. For example, please see the landmark study of Prieto et al 1993 (doi: 10.1109/10.532130), describing the variables that distinguish between healthy young and older adults. These include additional sway area and frequency measures. Also, it is not clear why the Anteroposterior measurements are not presented. One would expect that falling from a height should introduce instability to the AP component as well, due to the initial step-introduced momentum. 

We thank the review for this comment. The explanation of the postural variables now appears in the introduction (see the third and the fourth paragraphs.) 

A reminder sentence has also been added in section 2.4 Data Processing. This is the second last sentence.

More precisely :

  • The aim of the study is to assess the reliability of PLATES, with as specific interest for using it in the field for monitoring postural stability, and especially in the case of ankle instabilities.
  • To monitor ankle stability, the lateral axis is more sensitive than the anteroposterior axis, hence our focus on the lateral axis.
  • Thanks for pointing out the Prieto paper. We now quote it in the introduction (last sentence in the fourth paragraph)
  • We agree that falling from (a gentle) heigh induces an anteroposterior stabilization. Yet, in the SLL, the task is more difficult, and the instability is more visible on the lateral axis, mainly for mechanical reasons that relates to length-width ratio of the foot.

In the same vein, it is not clear why authors only present and discuss TSS and CoP velocity.

These two variables have already been used to predict ankle and knee injuries (Ross & al., 2011 ; Colby & al., 1999). Therefore, we have focused on these two parameters. But it is true that we should have presented and discussed the results of the other variables. The results of other parameters about the reliability are now presented in 3.2, 3.3, 3.4 in Results part, and discussed.

It is appreciated that authors provide evidence on the selection of the single leg tasks (lines 47-66), but double leg balance is also successful in identifying fallers and people with musculoskeletal diseases. Considering the relative danger of the single leg tasks for individuals with poor balance or suffering from ankle instability, further evidence/justification is needed to support the selection of the single leg balance tasks over the traditionally used double leg balance tasks (simple balance, Tandem stance, Romberg test etc). Please clearly state and support the task's physiological/biomechanical/functional importance.

Indeed, the two-legged balance can identify fallers and people suffering from musculoskeletal diseases, but as mentioned above, our validation study was carried out with the aim of assessing balance directly on the ground, and more specifically to prevent ankle sprains in sportswomen. As the participants were all sportswomen, we used static and dynamic unipodal tests, which have already been recommended to identify the risk of ankle sprain and/or prevent ankle instability (Trojian & McKeag, 2006; Wikstrom & al., 2005).

Next, we conducted the study only on women because they are poorly represented in scientific studies. In addition, women tend to be more fragile than men, especially with regard to ankle sprains, as they are 5.36 times more likely to be subject to ankle sprains than men (Engstrom & al., 1991).

The replication of the eyes closed/open result is an excellent idea. However, there are two points to be considered:

  1. It is mentioned at the line 105 for the first time, as coming out of the blue. I would suggest adding a rationale to support the utilization of this test alongside hypotheses in the introduction section.

A sentence explaining the value of testing with eyes open and closed was added to the introduction in the paragraph on Single Leg Balance (last sentence of the fourth paragraph)

  1. Other comparisons that are well established in the literature could be performed, such as young vs old adults (or people with neurological diseases), fallers vs non-fallers, or even dominant vs non dominant limb.

Thank you for this pertinent comment. We have therefore added, in the 3rd paragraph of the introduction, the different parameters that allow us to compare patients who fall vs. those who do not (Quijoux & al., 2020) and the different parameters that allow us to distinguish between young patients and adults (Prieto & al., 1996).

Minor comments:

Tables and figures:

Table 1: I am not sure if this table adds something to the paper in its current form. Please either expand it with more bipedal balance studies or delete. (Table 1 deleted)

Table S1 and S3: please correct the typo in the last column (“Biais” to “Bias”) (fixed)

Figures: Please explain the dashed lines (eg., 0, mean, +1 std, -1std, coef=1) in the caption (fixed)

Methods:

Line 104: please state that this forceplate was used as a gold standard reference. (fixed)

Lines 102-121: Please clarify what was randomised: tests, conditions, repetitions? (fixed, per condition, see 2.2 Experimental Design, the first sentences on SLB and SLL)

Lines 116, 117: It would be a good idea to write how many trials were excluded (fixed, see 2.2 Experimental Design, the last sentence of the second big paragraph)

Lines 120, 121: This statement is too vague. Did authors measure the variability? If yes, what was the criterion? We measured the variability with the variation coefficient. We added a table (table S4) in supplementary file with these results for each condition and each parameter. 

Reviewer 3 Report

The article  - “Validity and Reliability of Kinvent Plates for assessing single leg static and dynamic balance in the field” is performed at a good level and the article itself is written strictly in accordance with accepted rules. I would consider that this work can be published as it is. However, I consider it necessary to give one recommendation to the authors.

In my opinion, table 1 in the introduction is completely redundant. The experts included in the topic are well aware of what parameters are basic for evaluation of balance by using the Force platform. I think that it would be much more useful to cite works on the recommended standards for this type of study, which also contain recommended parameters. They have been discussed and published more than once. Some of the standards can be recognized as very old. However, the set of parameters for clinical analysis has remained virtually unchanged for decades. At least for the type of platforms used in the article - Force Platform. Such an approach, it seems to me, is much more useful for readers, and, perhaps, for authors as well.

In addition, the recorded and analyzed parameters are often selected based on the objectives of the study and may include quite complex data processing. However, for most clinical purposes, the parameters listed in your work are used.

Author Response

Thank you for your comments. You will find our response in bold. 

In my opinion, table 1 in the introduction is completely redundant. The experts included in the topic are well aware of what parameters are basic for evaluation of balance by using the Force platform. I think that it would be much more useful to cite works on the recommended standards for this type of study, which also contain recommended parameters. They have been discussed and published more than once. Some of the standards can be recognized as very old. However, the set of parameters for clinical analysis has remained virtually unchanged for decades. At least for the type of platforms used in the article - Force Platform. Such an approach, it seems to me, is much more useful for readers, and, perhaps, for authors as well.

In addition, the recorded and analyzed parameters are often selected based on the objectives of the study and may include quite complex data processing. However, for most clinical purposes, the parameters listed in your work are used.

We thank the review for this comment.

First of all, I would like to point out that the aim of the study is to assess the reliability of PLATES, with as specific interest for using it in the field for monitoring postural stability, and especially in the case of ankle instabilities.

We conducted the study only on women because they are poorly represented in scientific studies. In addition, women tend to be more fragile than men, especially with regard to ankle sprains, as they are 5.36 times more likely to be subject to ankle sprains than men (Engstrom & al., 1991).

The two-legged balance can identify fallers, people suffering from musculoskeletal diseases etc, but as mentioned above, this validation study was carried out with the aim of assessing balance directly on the ground, and more specifically to prevent ankle sprains in sportswomen. As the participants were all sportswomen, we used static and dynamic unipodal tests, which have already been recommended to identify the risk of ankle sprain and/or prevent ankle instability (Trojian & McKeag, 2006; Wikstrom & al., 2005). Consequently, I think it is therefore not necessarily useful to present standards for a bipodal balance exercise. But, in the introduction, we added sentences about the best parameters for assessing the balance, especially to assess the risk of falling in the elderly (Quijoux & al., 2020) and to characterize the difference in bipodal balance between young and older adults with eyes open and closed (Prieto & al., 1996).

The table 1 has been removed. We added the results of other parameters about the reliability in 3.2, 3.3, 3.4 in Results part, and discussed.

Reviewer 4 Report

This paper describes a preliminary validation study of a commercial force plate with a small population sample.

The limitation about the population sample and its features (geneder, age, ..) have been presented by the authors with good awareness.

the methodological approach is adequate but two points are missing:

1. the processing of the three repetitions is not clearly describes (do you chose one representative trial? do you compute an average of the computed parameters?

2. the COP trajectory for each trial could be analysed in more details with other parameters, but in particular the force values that are directly measured  have been completely ignored.

These data and related discussion should be included in the validation, that otherwise, remains incomplete.

Author Response

Thank you for your comments. You will find our response in bold. 

This paper describes a preliminary validation study of a commercial force plate with a small population sample.

The limitation about the population sample and its features (geneder, age, ..) have been presented by the authors with good awareness.

We thank the review for this comment.

First of all, I would like to point out that the aim of the study is to assess the reliability of PLATES, with as specific interest for using it in the field for monitoring postural stability, and especially in the case of ankle instabilities.

We conducted the study only on women because they are poorly represented in scientific studies. In addition, women tend to be more fragile than men, especially with regard to ankle sprains, as they are 5.36 times more likely to be subject to ankle sprains than men (Engstrom & al., 1991).

The two-legged balance can identify fallers, people suffering from musculoskeletal diseases etc, but as mentioned above, this validation study was carried out with the aim of assessing balance directly on the ground, and more specifically to prevent ankle sprains in sportswomen. As the participants were all sportswomen, we used static and dynamic unipodal tests, which have already been recommended to identify the risk of ankle sprain and/or prevent ankle instability (Trojian & McKeag, 2006; Wikstrom & al., 2005).

In the introduction, we added sentences about the best parameters for assessing the balance, especially to assess the risk of falling in the elderly (Quijoux & al., 2020) and to characterize the difference in bipodal balance between young and older adults with eyes open and closed (Prieto & al., 1996). The table 1 has been removed.

We added the results of other parameters about the reliability in 3.2, 3.3, 3.4 in Results part, and discussed.

the methodological approach is adequate but two points are missing:

  1. the processing of the three repetitions is not clearly describes (do you chose one representative trial? do you compute an average of the computed parameters?

We changed the description to make that clear in the method section “2.4 Data Processing” (last sentence of the first paragraph). 

  1. the COP trajectory for each trial could be analysed in more details with other parameters, but in particular the force values that are directly measured  have been completely ignored.

These data and related discussion should be included in the validation, that otherwise, remains incomplete.

We agree with you that many other important analyses could be have. For the present paper, we choose to focus other parameters than the forces, mainly because only record the vertical, force which is constant (the weight) for the SLB, hence not really informative.

In the SLL, in the laboratory, the medio-lateral and antero-posterior are useful to assess the stability with the time to stabilization parameter, but these forces are not recorded by the PLATES force platforms. Moreover, in the studies I have read about the SLL, the peak force on landing has never been analysed.

Round 2

Reviewer 1 Report

Thank you again for the opportunity to review the paper “Validity and Reliability of Kinvent Plates for assessing single leg static and dynamic balance in the field”.

All my comments and concerns have been addressed. I believe the paper can be now published in Sensors. Congratulations

Reviewer 4 Report

I appreciated the revised version receiving the suggested integrations.